# Effect of Gypsum, Compost, and Foliar Application of Some Nanoparticles in Improving Some Chemical and Physical Properties of Soil and the Yield and Water Productivity of Faba Beans in Salt-Affected Soils

Megahed M. Amer [1,*], Hesham M. Aboelsoud [1], Eman M. Sakher [1] and Ahmed A. Hashem [2,3,4]

1 Soil Improvement and Conservation Research Department, Soil, Water and Environment Research Institute (SWERI), Agriculture Research Center (ARC), Giza 12112, Egypt
2 College of Agriculture, Arkansas State University, Jonesboro, AR 72467, USA
3 Division of Agriculture, University of Arkansas System, Little Rock, AR 72204, USA
4 Agricultural Engineering Department, Suez Canal University, Ismailia 41522, Egypt
* Correspondence: megahedamer@arc.sci.eg

**Abstract:** Two field experiments were conducted at Kafr El Sheikh Gov., Egypt, during two winter growing seasons (2020/2021 and 2021/2022). The objective of this study was to improve some chemical and physical properties of soil and the yield and water productivity of faba beans (*Viciafaba* L.), Cv. Sakha-4 by application of gypsum, compost, and some nanoparticles in salt-affected soils. The experimental treatments were arranged in a split-plot design with three replications. The main plots had the following soil amendments: $T_1$: control treatment, $T_2$: 10 tons compost/hectare, $T_3$: soil gypsum requirement (GR) of 8.59 ton ha$^{-1}$, and $T_4$: GR + 10 tons compost/hectare. The subplots were treated with foliar application as follows: no treatment, manganese nanoparticles (Mn-NPs), selenium nanoparticles (Se-NPs), and Mn-NPs + Se-NPs. According to the findings, the application of compost + GR significantly decreased soil salinity (EC), exchangeable sodium percentage (ESP), and soil bulk density (BD). However soil porosity, soil penetration resistance (SPRa), and basic soil infiltration (IR) were significantly increased. On the other hand, the results revealed significant positive effects onthe 100-grain weight as well as proline, chlorophyll, superoxide dismutase, and catalase contents due to the interaction between gypsum + compost and Mn-NPs + Se-NPs, which enhanced the productivity of both the seed and straw yields of faba beans compared to the alternative treatments. In addition, the seed yield and irrigation water productivity (PIW, kg m$^3$) of faba beans were significantly increased with addition of gypsum and compost and foliar application of nanoparticles. The highest values of these parameters were achieved due to the interaction between gypsum + compost and Mn-NPs + Se-NPs. It can be concluded that application of GR of 8.59 ton ha$^{-1}$ and 10 ton ha$^{-1}$ compost as well as foliar application of Mn-NPs and Se-NPs may be a key strategy for improving some chemical and physical properties of soil and the yield and water productivity of faba beans in salt-affected soil under these experimental conditions.

**Keywords:** bulk density; soil infiltration; exchangeable sodium percentage; proline; catalase; faba bean; irrigation water productivity

## 1. Introduction

Salt-affected soils in the Nile Delta may result from low rainfall and high evapotranspiration rates, low irrigation water quality, water logging, and saline sea water intrusion, which cause a decline in productivity [1]. The management of salt-affected soils is mainly focused on decreasing soil salinity by leaching the affiliated ions out of the soil using materials such as gypsum and high-quality water [2] and the application of compost as a soil amendment [3]. The adsorption of Na$^+$ on the exchangeable sites of clay particles is

considered responsible for soil dispersion. Gypsum can prevent soil dispersion by maintaining high Ca/Na ratios, thus promoting clay flocculation and structural stability [2,4,5]. Meanwhile, applying compost as a soil amendment has been recognized as a reliable way to rebuild the physicochemical properties of soil and re-establish the microbial populations and activities in soils, especially those with poor structure and low organic matter [6,7]. The addition of compost to soil can alleviate the negative impacts of salinity via modulation of key plant functions, including growth, nutrient uptake, and upregulation of the ascorbate–glutathione cycle, which subsequently affects the plant yield [8–11]. The application of compost enhances soil water-holding capacity and the infiltration rate of saline–sodic soils and also increases soil nutrient and organic matter content [12]. As for plants, applying compost increases the yield, yield components, and total crude protein of faba beans [13,14]. The combination of sulfur and organic amendments can improve soil characteristics such as salinity, organic carbon, and available NPK [15–17] and increase faba bean pods and seed yield as well as the nitrogen, phosphor, potassium, and zinc contents in seeds [3,18]. Nanofertilizers have been found to enhance the capability of plants to absorb nutrients from the soil [19], therefore enhancing plant growth and tolerance of plants to biotic and abiotic stresses and increasing the yield and plant biomass [20–22]. In addition, foliar application of nanofertilizers has been found to be appropriate for field use because it gradually feeds plants in a controlled manner compared to salt fertilizers [23] while controlling the toxicity that may occur after soil amendment with the same nutrients [24]. Among nanoparticles, silica (Si-NPs) and manganese (Mn-NPs) can significantly alleviate the effect of salt stress on the growth of *Arabidopsis thaliana* [25], cucumber [26], and grapevine [27].

Faba bean (*Viciafaba* L.) is a source of protein that provides a renewable nitrogen input to crops and soils [28]. The Egyptian Government is pressing hard to increase the yield and quality of faba bean crops through agricultural practices such as soil amendments and foliar application of some nanoparticles.

The application of nanofertilizers has been investigated as a way to support global food production. Nanofertilizers have positive impacts under stressful growth conditions, such as those found in salt-affected soils, because they are slow-releasing fertilizers with high nutrient use efficiency [29]. The role of nutrient nanoparticles under stress has been investigated for crops such as rice [29], faba beans [3], and bananas [30]. In addition to nanofertilizers, halophytic-based nanoparticles have been shown to improve crop productivity under salinity stress by improving water use efficiency and enhancing the ion flux, plant photosynthesis efficiency, production of proteins involved in oxidation–reduction reactions, and hormonal signaling pathways [31,32]. Research into crop production on salt-affected soils remains an extremely important topic that needs more attention and funding [33]. This study aimed to evaluate the following aspects:

1. Soil application of gypsum, compost, and gypsum + compost;
2. Foliar application of Mn-NPs, Se-NPs, and Mn-NPs + Se-NPs;
3. Interaction effects between soil amendments and foliar application of nanoparticles on
   (a) improving some chemical and physical properties of soil;
   (b) alleviation of the negative effects of salinity on growth and grain and straw yields of faba bean plants;
   (c) productivity of irrigation water.

## 2. Materials and Methods

### 2.1. Description of the Location

Two field experiments were conducted in, Kafr El Sheikh Gov., Egypt (which lies between 31°19′03.3″ Nand 31°01′24.3″ E), during two winter growing seasons (2020/2021 and 2021/2022) to study the effects of the application of gypsum($CaSO_4 \cdot 2H_2O$), compost, and some nanomaterialsonsome chemical and physical properties of soil as well as the yield and water productivity of faba beans in salt-affected soils.

## 2.2. Experimental Design

The experimental treatments were arranged in a split-plot design with three replications. The main plots had the following soil amendments: $T_1$: control treatment, $T_2$: 10 tons compost/hectare, $T_3$: soil gypsum requirement (GR), and $T_4$: GR + 10 tons compost/hectare. The subplots were treated with foliar application as follows: no treatment, manganese nanoparticles (Mn-NPs), selenium nanoparticles (Se-NPs), and Mn-NPs + Se-NPs. Therefore, the experimental units consisted of 48 plots (4 soil amendments $\times$ 4NPs $\times$ 3 rep.), and the area of each plot was 42 $m^2$ (6 $\times$ 7 m). Faba beans (Cv. Sakha-4) weresown on 20 October and 25 October for both the 2020/2021 and 2021/2022 seasons.

## 2.3. Materials Used and Their Source

All the recommended agronomic practices were applied. N-fertilizer was applied at a rate of 15 kg fed$^{-1}$ as a starter application. The recommended phosphorus (P) application rate of 15 Kg $P_2O_5$ fed$^{-1}$ was added as a monophosphate (15.5% $P_2O_5$), and the recommended potassium (K) application of 30 Kg $K_2O$ fed$^{-1}$ was added as potassium sulfate (48% $K_2O$) before tillage. The compost was made from a mixture of residual plants and animals, and its chemical composition was as follows: N: 1.45 mg kg$^{-1}$, P: 0.67 mg kg$^{-1}$, K: 2.19 mg kg$^{-1}$, organic matter: 37.9%, C/N ratio: 19:1, pH: 7.69, EC: 2.71 dS m$^{-1}$, bulk density: 812 kg m$^{-3}$, and moisture content: 28.21%.

The size of Se-NPs was further confirmed by transmission electron microscopy (TEM) imaging, which demonstrated that the Se particles possessed an average diameter of 40–80 nm. The zeta potential measurements indicated a high negative charge ($-45.16$ mV). Senanoparticles were prepared biologically using Bacillus cereus strain culture as the bacteria strain from the Agricultural Microbiology Department, Soils, Water and Environment Research Institute (SWERI), Agricultural Research Center (ARC), according to [34].

The Mn-NPs were provided by the National Research Center (NRC), Egypt. The specific surface area and bulk density of $Mn_2O_3$ NPs were estimated to be 155 $m^2$/g and 0.36 g/cm$^3$, respectively. The zeta potential of the particle was low and had a negative value ($-5.8$ mV at pH 7). The spherical-shaped $Mn_2O_3$ NPs were approximately 35 nm in size. Both Se-NPs [35] and Mn-NPs [36] were applied at a rate of 100 mg L$^{-1}$. Each nanoparticle was applied two times with two-week intervals starting 25 days after the sowing of faba beans.

## 2.4. Soil Sampling and Laboratory Analysis

Soil samples were collected from 3 consecutive depths (0–20, 20–40, and 40–60 cm) before experiments and after harvesting for all treatments to carry out physical and chemical analysis. The salinity of the saturated soil paste extract ($EC_e$), cation exchange capacity (CEC), and exchangeable sodium percentage (ESP) were determined according to [37]. Organic matter (OM) content was determined according to [38]. Soil bulk density and total porosity of the different soil layers for all treatments were measured before experiments and after harvesting using the core sampling technique described by [39]. Particle size distribution of soil was measured using the pipette method according to [40]. Infiltration rate was determined using a double-cylinder infiltrometer as described by [41]. Field capacity (FC) and wilting point (WP) were determined using the pressure membrane method at 0.33 and 15 bars, respectively [42]. Gypsum requirement (GR) was determined according to [43]. To reduce the initial soil ESP to the desired ESP (10) in the surface layer (0–30 cm), the following equation was applied:

$$GR = (ESP_i - ESP_f)/100 \times CEC \times 1.72 \times (100/\text{purity})$$

$$= (16.21 - 10)/100 \times 31.84 \times 1.72 \times (100/95) \tag{1}$$

$$= 8.59 \text{ ton/ha}^{-1}$$

where GR: gypsum requirement (ton ha$^{-1}$) for the upper 30 cm soil, ESP$_i$: initial soil ESP, ESP$_f$: the desired soil ESP and CEC: cation exchange capacity (cmolc kg$^{-1}$), and 1.72 is the amount of CaSO$_4$·2H$_2$O (ton) required to reduce Na$^+$ content of the soil by one unit (1 cmolc Na 100 g$^{-1}$ soil). Compost was applied as recommended for this area by [44]. The gypsum requirement and compost were added before soil tillage. The experimental soil's physical and chemical properties are shown in Table 1. The meteorological data from the Sakha Station during the growing seasons are presented in Table 2.

**Table 1.** Some physical and chemical properties of the soil before the experiment.

| Soil (cm) | pH | CaCO$_3$ (%) | EC (dS m$^{-1}$) | ESP (%) | Available Macronutrients (mg kg$^{-1}$) | | | OM (%) | CEC (cmole kg$^{-1}$) |
|---|---|---|---|---|---|---|---|---|---|
| | | | | | N | P | K | | |
| 0–20 | 8.01 ± 0.01 | 1.88 ± 0.1 | 10.15 ± 0.21 | 15.87 ± 0.35 | 18 ± 0.61 | 10 ± 0.45 | 254 ± 2.10 | 1.70 ± 0.01 | 32.55 ± 0.12 |
| 20–40 | 8.15 ± 0.01 | 2.10 ± 0.11 | 11.12 ± 0.19 | 16.55 ± 0.41 | 16 ± 0.87 | 9 ± 0.51 | 251 ± 2.50 | 1.65 ± 0.02 | 31.14 ± 0.13 |
| 40–60 | 8.16 ± 0.02 | 2.19 ± 0.12 | 11.91 ± 0.12 | 18.91 ± 0.42 | 16 ± 0.98 | 7 ± 0.55 | 249 ± 2.45 | 1.51 ± 0.01 | 30.45 ± 0.21 |
| | FC (%) | WP (%) | AW (%) | BD (kg m$^{-3}$) | | IR (cm/h) | PR (N cm$^{-2}$) | | |
| 0–20 | 45.11 ± 0.21 | 22.01 ± 0.15 | 23.10 ± 0.19 | 1.31 ± 0.01 | | 0.59 ± 0.12 | 280.14 ± 7.15 | | |
| 20–40 | 42.52 ± 0.23 | 20.28 ± 0.18 | 22.24 ± 0.14 | 1.32 ± 0.01 | | | 287.51 ± 6.45 | | |
| 40–60 | 40.50 ± 0.14 | 19.03 ± 0.21 | 21.47 ± 0.12 | 1.32 ± 0.01 | | | 291.42 ± 6.98 | | |
| | Sand (%) | Silt (%) | Clay (%) | Soil texture | | | | | |
| 0–20 | 15.55 ± 0.21 | 26.25 ± 0.45 | 58.20 ± 0.61 | Clay | | | | | |
| 20–40 | 18.55 ± 0.14 | 24.95 ± 0.65 | 56.50 ± 0.71 | Clay | | | | | |
| 40–60 | 18.82 ± 0.15 | 25.23 ± 0.61 | 55.95 ± 0.56 | Clay | | | | | |

pH: pH of soil/water suspension (1:2.5); EC: electrical conductivity; ESP: exchangeable sodium percentage; CEC: cation exchange capacity; OM: organic matter; CEC: cation exchange capacity, FC: field capacity; WP: wilting point; AW: available water; BD: bulk density; IR: soil basic infiltration rate; PR: penetration resistance. Values are means ± standard deviation (SD) from three replicates (means ± SD).

**Table 2.** Meteorological data during the 2020/2021 and 2021/2021 growing seasons.

| | Temperature (°C) | | | | Wind Speed (km Day$^{-1}$) | | Relative Humidity, % | | Rainfall (mm Month$^{-1}$) | |
|---|---|---|---|---|---|---|---|---|---|---|
| | 2020/2021 | | 2021/2022 | | 2020/2021 | 2021/2022 | 2020/2021 | 2021/2022 | 2020/2021 | 2021/2022 |
| | Max | Min | Max | Min | | | | | | |
| Oct. | 31.5 | 24.6 | 29.41 | 21.50 | 72.7 | 80.23 | 60.19 | 68.87 | - | - |
| Nov. | 25.0 | 17.5 | 26.64 | 18.84 | 46.9 | 64.73 | 66.42 | 72.25 | 18.35 | 12.70 |
| Dec. | 22.9 | 13.7 | 20.15 | 15.05 | 44.9 | 62.81 | 67.66 | 74.11 | 18.78 | 25.07 |
| Jan. | 21.0 | 13.5 | 17.84 | 9.88 | 39.2 | 62.35 | 68.14 | 75.55 | 14.05 | 50.35 |
| Feb. | 21.5 | 12.5 | 19.22 | 10.77 | 58.3 | 79.75 | 68.36 | 70.75 | - | 25.25 |
| Mars | 23.8 | 15.2 | 19.17 | 14.15 | 83.4 | 98.45 | 67.11 | 69.01 | 5.40 | 5.25 |
| Apr. | 27.6 | 19.4 | 27.64 | 19.74 | 95.0 | 138.67 | 60.32 | 60.95 | - | - |

*2.5. Yield*

Harvesting was carried out 160 days after sowing at a moisture level of 15.5% from each plot to calculate yield, including the 100-grain weight, grain and straw yields, and productivity of irrigation water (PIW; kg m$^{-3}$). PIW is a quantitative term used to define the relationship between crop produced and the amount of water involved in crop production, and it can be calculated according to [45] as follows:

$$\text{PIW} = \text{Grain yield (kg ha}^{-1}\text{)}/\text{water applied (m}^3\text{ ha}^{-1}\text{)} \qquad (2)$$

A frozen sample of 0.5 g of 45-day-old plant samples (using a fully extended upper leaf devoid of the midribs) was homogenized in 8 mL of 50 mM cold phosphate buffer at pH 7 (modified from [46]). The homogenates were centrifuged at $4000 \times g$ rpm for 20 min, and the supernatant was used as a crude extract for enzymatic assay. Catalase (CAT) activity was determined as a decrease in absorbance at 240 nm for 1 min following the decomposition of H$_2$O$_2$ [47]. Superoxide dismutase (SOD; µg$^{-1}$ FW) was assayed based on its ability to inhibit the photochemical reduction of nitro blue tetrazolium [46].

### 2.6. Photosynthetic Pigments

Chlorophyll (Chl) is a photosynthetic pigment that absorbs solar energy for photosynthesis and is susceptible to various environmental conditions. The total chlorophyll in tissues taken from the second completely developed leaf at the plant's tip was measured 60 days after the grains were sown. The content of the photosynthetic pigments was calculated according to [48]. In brief, 0.1 g of fresh leaf tissue was ground in 5 mL of acetone 80%, followed by a 10 min centrifugation process at $13,000\times g$ rpm. Using a UV spectrophotometer (Model 6705, UK), the supernatant's absorbance was measured at 645, 663, and 470 nm to determine the extract's chlorophyll (mg g$^{-1}$ FW)

### 2.7. Proline Content

The endogenous proline content in the second completely grown leaf from the plant tip was measured 60 days after the date of sowing [49]. In brief, 0.1 g of fresh plant tissues was combined with 4 mL of 3.0% sulfosalicylic acid in a mortar and stored at 5 °C overnight. The suspension was centrifuged at room temperature for 5 min at $3000\times g$ rpm. With the supernatant, 4 mL of acidic ninhydrin reagent was mixed. After being mechanically shaken, tubes were heated in a boiling water bath for one hour. The mixture was then extracted with 4 mL of toluene in a separating funnel once the tubes were cooled. Using spectrophotometry, the absorbance of the toluene layer was measured at 520 nm. Regarding the standard curve, the concentration of the unidentified sample was estimated. Six samples on average were used for each treatment in the final value.

### 2.8. Statistical Analysis

The data were analyzed statistically by analysis of variance (ANOVA) using the Co-State program, version 6.303, according to [50]. Treatment means were compared by Duncan's multiple range test at 0.05 and 0.01 levels of significance [6].

## 3. Results and Discussion

### 3.1. Chemical Properties of Soil

As shown in Table 3, the salinity of the soil as measured by electrical conductivity (EC$_e$) highly significantly decreased with the application of compost and gypsum. The lowest values (6.86 and 4.72 dS m$^{-1}$)for the 2020/2021 and 2021/2022 seasons were recorded with the application of compost + gypsum. The same trend was seen with exchangeable sodium percentage (ESP), which highly significantly decreased with the application of soil amendments. The lowest values were recorded with the application of compost + gypsum. The impacts were in the following descending order: compost + gypsum > gypsum > compost > control treatment in both growing seasons. The use of gypsum and compost decreased soil salinity by leaching the affiliated ions out of the soil. The application of gypsum can prevent soil dispersion by maintaining high Ca/Na ratios, thus promoting clay flocculation and structural stability. These results are supported by [2,3].

The chemical properties of soil, such as EC$_e$ and ESP, were not significantly affected by foliar application of Mn-NPsand Se-NPs. EC$_e$ was significantly decreased and recorded the lowest values (6.8 dS m$^{-1}$ and 4.66 dS m$^{-1}$) for the two growing seasonsdue to the interaction between application of compost + gypsum and Mn-NPs + Se-NPs after harvesting of faba beans. ESP showed the same trend and recorded the lowest values (10.97 and 10.34%) due to the interaction between application of compost + gypsum and foliar application of Mn-NPs + Se-NPs after harvesting of faba beans for the 2020/2021 and 2021/2022 growing seasons. These results are supported by [2,17], which showed that the combination between gypsum and organic amendments could improve the chemical properties of soil.

**Table 3.** Effect of gypsum, compost, and foliar application of some nanomaterials on electrical conductivity (EC$_e$) and exchangeable sodium percentage (ESP) during the 2020/2021 and 2021/2022 seasons.

| Treatments | | EC (dS m$^{-1}$) | | ESP (%) | |
|---|---|---|---|---|---|
| **A** | **B** | **2020/2021** | **2021/2022** | **2020/2021** | **2021/2022** |
| Control | without | 9.63 ± 0.40 [ab] | 9.08 ± 0.04 [ab] | 15.04 ± 0.03 [a] | 14.86 ± 0.01 [a] |
| | Mn-NPs | 9.64 ± 0.01 [a] | 9.09 ± 0.01 [a] | 15.05 ± 0.01 [ab] | 14.87 ± 0.0 [a] |
| | Se-NPs | 9.63 ± 0.02 [ab] | 9.08 ± 0.02 [ab] | 15.04 ± 0.01 [ab] | 14.86 ± 0.02 [a] |
| | Mn + Se | 9.61 ± 0.01 [bc] | 9.06 ± 0.01 [bc] | 15.03 ± 0.01 [bc] | 14.65 ± 0.0 [b] |
| Compost | without | 9.61 ± 0.01 [bc] | 9.06 ± 0.01 [bc] | 15.03 ± 0.01 [c] | 14.65 ± 0.0 [b] |
| | Mn-NPs | 9.60 ± 0.01 [bc] | 9.06 ± 0.01 [c] | 15.02 ± 0.01 [bc] | 14.64 ± 0.0 [b] |
| | Se-NPs | 9.60 ± 0.01 [bc] | 9.05 ± 0.01 [c] | 15.02 ± 0.01 [bc] | 14.64 ± 0.1 [b] |
| | Mn + Se | 9.61 ± 0.01 [bc] | 9.05 ± 0.01 [c] | 15.03 ± 0.01 [c] | 14.65 ± 0.0 [b] |
| Gypsum | without | 7.62 ± 0.02 [d] | 6.64 ± 0.02 [d] | 12.06 ± 0.02 [d] | 11.89 ± 0.01 [c] |
| | Mn-NPs | 7.62 ± 0.02 [d] | 6.64 ± 0.02 [d] | 12.06 ± 0.02 [d] | 11.89 ± 0.01 [c] |
| | Se-NPs | 7.60 ± 0.02 [d] | 6.62 ± 0.02 [e] | 11.86 ± 0.01 [e] | 11.88 ± 0.0 [c] |
| | Mn + Se | 7.57 ± 0.01 [e] | 6.59 ± 0.01 [f] | 11.84 ± 0.01 [e] | 11.86 ± 0.01 [d] |
| C + G | without | 6.91 ± 0.02 [f] | 4.77 ± 0.02 [g] | 11.18 ± 0.01 [f] | 10.38 ± 0.02 [e] |
| | Mn-NPs | 6.90 ± 0.01 [f] | 4.76 ± 0.01 [g] | 11.12 ± 0.00 [f] | 10.38 ± 0.02 [e] |
| | Se-NPs | 6.83 ± 0.01 [g] | 4.69 ± 0.01 [h] | 10.99 ± 0.00 [g] | 10.36 ± 0.02 [f] |
| | Mn + Se | 6.80 ± 0.02 [h] | 4.66 ± 0.02 [i] | 10.97 ± 0.01 [g] | 10.34 ± 0.00 [g] |
| Main plot (A) | LSD$_{0.05}$ | 0.006 | 0.006 | 0.005 | 0.007 |
| | LSD$_{0.01}$ | 0.009 | 0.009 | 0.008 | 0.012 |
| Sub plot (B) | LSD$_{0.05}$ | 0.008 | 0.008 | 0.006 | 0.005 |
| | LSD$_{0.01}$ | 0.01 | 0.01 | 0.008 | 0.006 |
| Interaction (A × B) | LSD$_{0.05}$ | 0.01 | 0.016 | 0.012 | 0.009 |
| | LSD$_{0.01}$ | 0.02 | 0.021 | 0.012 | 0.013 |

Means followed by different letters indicate significant differences among treatments according to the Duncan's test ($p < 0.01$). Values are means ± standard deviation (SD) from three replicates (means ± SD). The values are for soil collected at depths of 0–20, 20–40, and 40–60 cm.

### 3.2. Physical Properties of Soil

As shown in Figure 1, the soil bulk density significantly decreased with the application of soil amendments. The lowest values (1.308 and 1.303 kg m$^{-3}$) were recorded with the application of compost + gypsum for the 2020/2021 and 2021/2022 growing seasons compared to the other treatments. The impacts of different treatments were in the following order: compost + gypsum > gypsum > compost > control treatment (Figure 1A).

As shown in Figure 1B, the soil porosity was significantly increased by the application of soil amendments, and the highest values (50.64 and 50.83%) were recorded for application of compost + gypsum for the 2020/2021 and 2021/2022 growing seasons. Soil penetration resistance was significantly decreased with the application of compost or gypsum and recorded the lowest values (245.57 and 230.44 N cm$^{-2}$) with compost + gypsum for the 2020/2021 and 2021/2022 growing seasons. These results are supported by the authors of [6,7], who reported that applying compost is recognized as a reliable way to rebuild the physicochemical properties of soil and promote microbial activities in the soil, especially in soils with poor structure and low organic matter.

Soil basic infiltration rate showed the same trend, and the highest values (1.25 and 1.32 cm h$^{-1}$) were recorded for soil application of compost + gypsum for the 2020/2021 and 2021/2022 growing seasons (Figure 2). These results may be attributed to the application of compost to saline–sodic soils, which enhances its infiltration rate according to [12]. Organic ameliorators have many benefits for soil health as they enhance physical (soil porosity and bulk density) and chemical (EC and ESP) soil quality parameters [51].

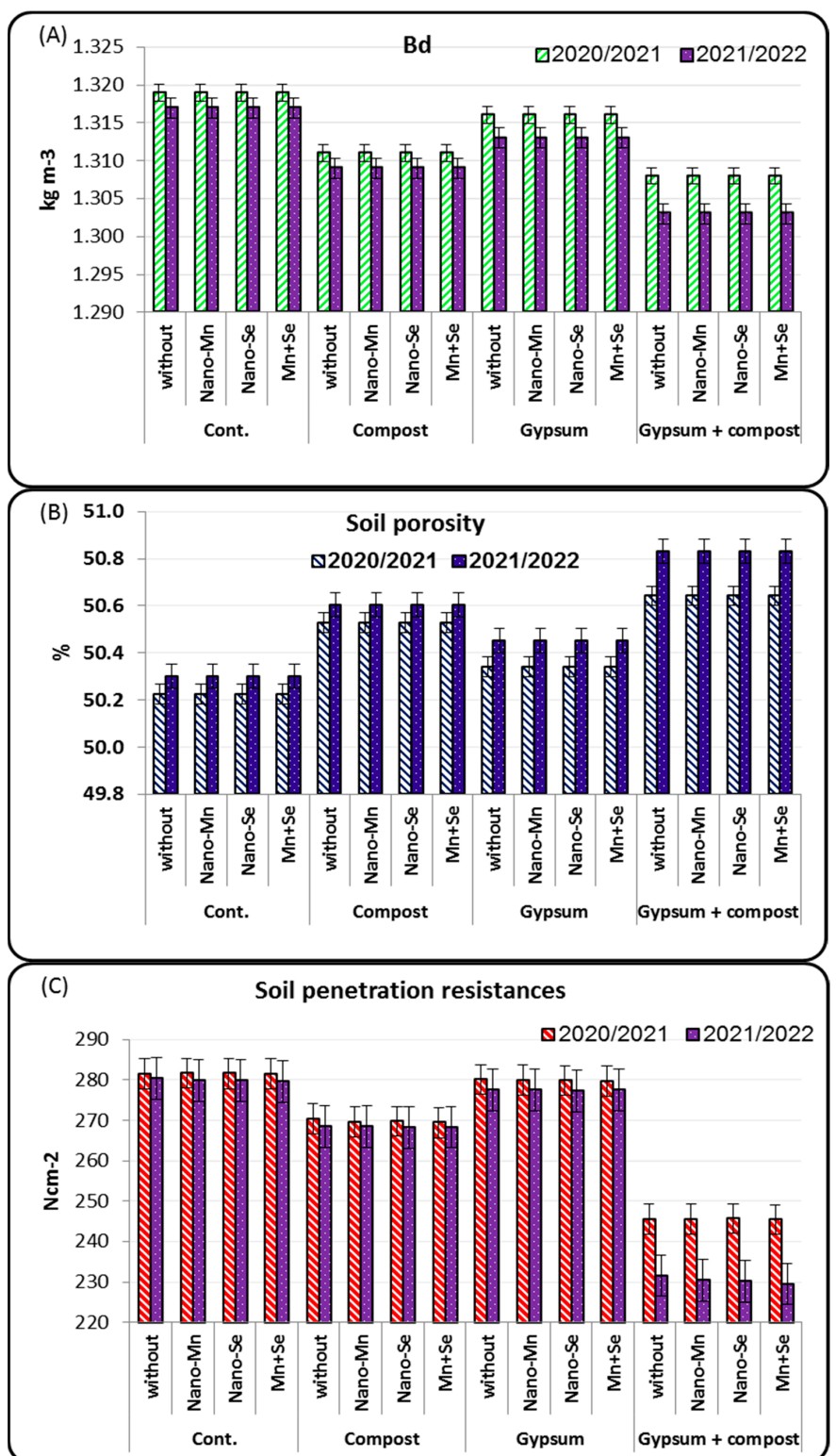

**Figure 1.** Effect of compost, gypsum, and foliar application of some nanoparticles on soil bulk density (**A**), soil porosity (**B**), and soil penetration resistance (SPRa) (**C**) in the 2020/2021 and 2021/2022 seasons. The values are for soil collected at depths of 0–20, 20–40, and 40–60 cm.

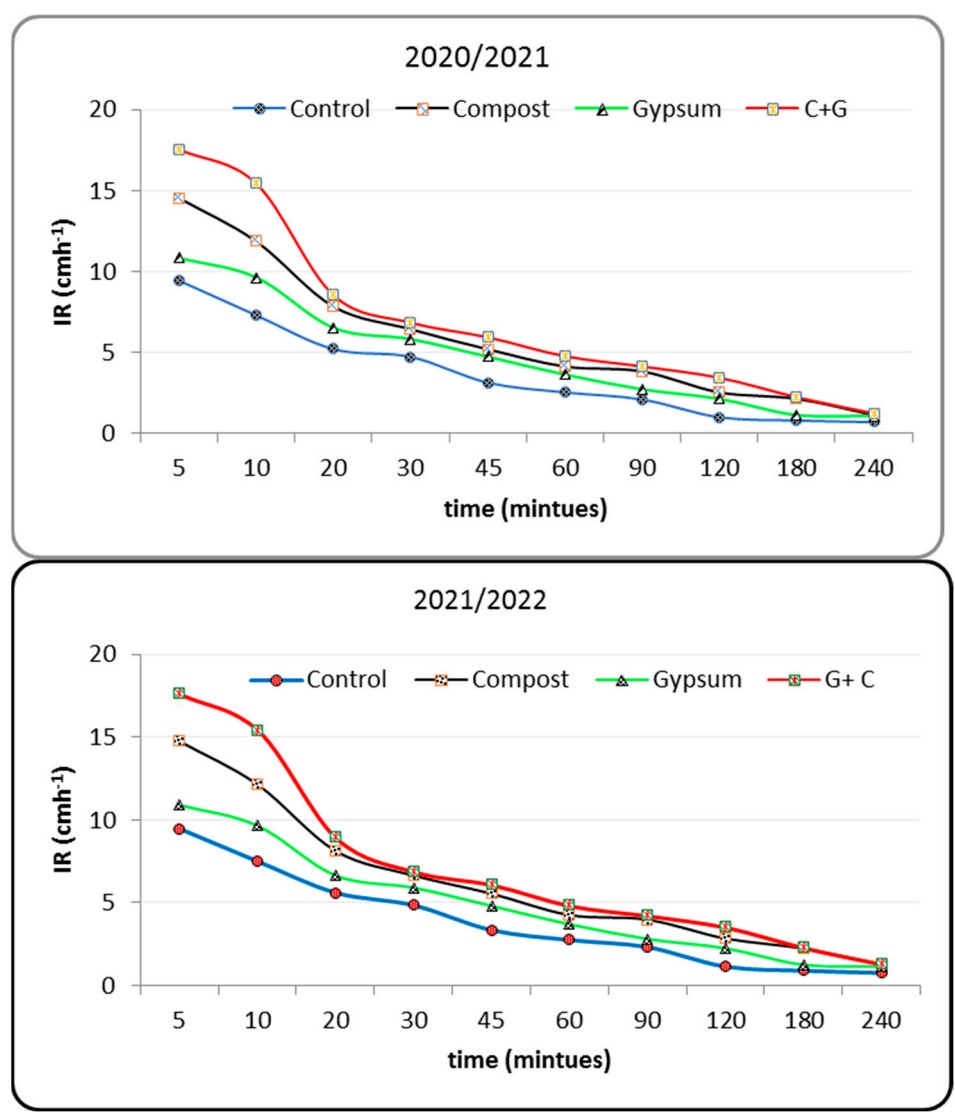

**Figure 2.** Effect of soil amendments of gypsum and compost on soil basic infiltration rate (IR) in the 2020/2021 and 2021/2022 seasons.

### 3.3. The 100-Grain Weight, Proline and Chlorophyll Contents of Faba Bean

As shown in Table 4, the 100-grain weight was significantly increased by applying compost and/or gypsum and the highest values (69.62 and 69.73 gm) were recorded with the application of compost + gypsum for the 2020/2021 and 2021/2022 growing seasons. These results may be related to the enhancement effects of compost on plant growth as reported by [13,14].

The data showed that the 100-grain weight was highly significantly increased with foliar application of Mn-NPs and/or Se-NPs and, the highest values (66.79 and 67.03 g) for the first and second growing seasons were recorded with the application of Mn-NPs + Se-NPs. Moreover, the 100-grain weight was significantly increased due to the interaction between all treatments, and the highest values (70.56 and 70.77 g) for the 2020/2021 and 2021/2022 growingseasons were achieved by applying compost + gypsum as well as foliar application of Mn-NPs + Se-NPs.

The proline content in the plant was highly significantly increased by soil application of compost and gypsum, and the highest values (3.83 and 3.82 $\mu$mol g$^{-1}$ FW) were recorded with the application of compost + gypsumin both the growing seasons. In addition, the obtained data cleared showed that the proline content in plants was highly significantly increased with foliar application of Mn-NPs and/or Se-NPs, and the highest values (3.59

and 3.58 μmol g$^{-1}$ FW) for both the 2020/2021 and 2021/2022 growing seasons were recorded with the foliar application of Mn-NPs + Se-NPs. In addition, the proline content was significantly increased due to the interaction between all treatment under study, and the highest values (3.86 and 3.85 μmol g$^{-1}$ FW) were obtained with the compost + gypsum with foliar application of Mn-NPs + Se-NPs for both seasons (Table 4). This positive trend may be attributed to the fact that nanoparticles of silica and manganese alleviate theinhibitory effects of salt stress on plant growth, as shown by [27].

**Table 4.** Combined effects of soil amendments and foliar application of some nanoparticles on 100-grain weight and proline and chlorophyll contents (mg g$^{-1}$ FW) in faba beans for the 2020/2021 and 2021/2022growing seasons.

| Treat. | | 100-Grain Weight (g) | | Proline (μmol g$^{-1}$ FW) | | Chlorophyll (mg g$^{-1}$ FW) | |
|---|---|---|---|---|---|---|---|
| A | B | 2021 | 2022 | 2021 | 2022 | 2021 | 2022 |
| Control | without | 59.17 ± 0.13 [l] | 59.27 ± 0.07 [m] | 2.56 ± 0.03 [m] | 2.50 ± 0.02 [n] | 0.43 ± 0.01 [k] | 0.45 ± 0.01 [l] |
| | Mn-NPs | 61.23 ± 0.24 [k] | 61.46 ± 0.42 [l] | 2.64 ± 0.02 [l] | 2.60 ± 0.00 [m] | 0.45 ± 0.03 [k] | 0.47 ± 0.03 [l] |
| | Se-NPs | 61.24 ± 0.21 [k] | 61.51 ± 0.30 [kl] | 2.88 ± 0.02 [k] | 2.86 ± 0.01 [l] | 0.49 ± 0.0 [j] | 0.51 ± 0.00 [k] |
| | Mn + Se | 61.63 ± 0.16 [j] | 61.74 ± 0.20 [jk] | 3.11 ± 0.01 [j] | 3.10 ± 0.01 [k] | 0.49 ± 0.01 [j] | 0.51 ± 0.01 [k] |
| Compost | without | 61.48 ± 0.11 [j] | 61.63 ± 0.11 [j] | 3.59 ± 0.02 [f] | 3.58 ± 0.02 [g] | 0.53 ± 0.03 [i] | 0.55 ± 0.03 [j] |
| | Mn-NPs | 66.23 ± 0.26 [i] | 66.29 ± 0.14 [j] | 3.65 ± 0.02 [e] | 3.64 ± 0.01 [f] | 0.58 ± 0.03 [h] | 0.60 ± 0.03 [i] |
| | Se-NPs | 66.70 ± 0.87 [h] | 67.08 ± 0.05 [i] | 3.67 ± 0.01 [d] | 3.66 ± 0.01 [e] | 0.63 ± 0.00 [g] | 0.65 ± 0.00 [h] |
| | Mn + Se | 66.95 ± 0.53 [g] | 67.45 ± 0.19 [h] | 3.70 ± 0.01 [c] | 3.69 ± 0.02 [d] | 0.66 ± 0.01 [f] | 0.68 ± 0.01 [g] |
| Gypsum | without | 61.93 ± 0.205 [j] | 61.74 ± 0.18 [g] | 3.23 ± 0.01 [i] | 3.21 ± 0.01 [j] | 0.70 ± 0.04 [e] | 0.72 ± 0.04 [f] |
| | Mn-NPs | 67.63 ± 0.20 [f] | 67.79 ± 0.04 [f] | 3.28 ± 0.02 [h] | 3.27 ± 0.01 [i] | 0.76 ± 0.03 [d] | 0.78 ± 0.03 [e] |
| | Se-NPs | 67.81 ± 0.07 [e] | 67.88 ± 0.06 [ef] | 3.34 ± 0.01 [g] | 3.33 ± 0.01 [h] | 0.80 ± 0.02 [c] | 0.82 ± 0.02 [d] |
| | Mn + Se | 68.02 ± 0.12 [de] | 68.16 ± 0.09 [e] | 3.68 ± 0.02 [cd] | 3.67 ± 0.01 [e] | 0.82 ± 0.02 [c] | 0.84 ± 0.02 [cd] |
| C + G | without | 67.88 ± 0.05 [d] | 67.96 ± 0.04 [d] | 3.81 ± 0.01 [b] | 3.79 ± 0.01 [c] | 0.82 ± 0.02 [c] | 0.84 ± 0.02 [c] |
| | Mn-NPs | 69.85 ± 0.10 [c] | 69.91 ± 0.08 [c] | 3.81 ± 0.01 [b] | 3.80 ± 0.01 [c] | 0.89 ± 0.02 [b] | 0.91 ± 0.02 [b] |
| | Se-NPs | 70.20 ± 0.17 [b] | 70.32 ± 0.22 [b] | 3.83 ± 0.01 [b] | 3.82 ± 0.01 [b] | 0.90 ± 0.02 [b] | 0.92 ± 0.02 [b] |
| | Mn + Se | 70.56 ± 0.15 [a] | 70.77 ± 0.21 [a] | 3.86 ± 0.02 [a] | 3.85 ± 0.02 [a] | 0.93 ± 0.02 [a] | 0.95 ± 0.02 [a] |
| Main plot (A) | LSD$_{0.05}$ | 0.175 | 0.139 | 0.005 | 0.005 | 0.013 | 0.014 |
| | LSD$_{0.01}$ | 0.266 | 0.211 | 0.008 | 0.008 | 0.02 | 0.021 |
| Sub plot (B) | LSD$_{0.05}$ | 0.079 | 0.067 | 0.007 | 0.004 | 0.008 | 0.009 |
| | LSD$_{0.01}$ | 0.107 | 0.091 | 0.009 | 0.006 | 0.011 | 0.011 |
| Interaction (A × B) | LSD$_{0.05}$ | 0.158 | 0.135 | 0.014 | 0.001 | 0.017 | 0.018 |
| | LSD$_{0.01}$ | 0.215 | 0.183 | 0.019 | 0.008 | 0.023 | 0.023 |

Means followed by different letters indicate significant differences among treatments according to the Duncan's test ($p < 0.01$). Values are means ± standard deviation (SD) from three replicates (means ± SD).

Chlorophyll content in plant leaves was significantly increased by applying compost and gypsum or their combination. The highest values (0.887 and 0.907 mg g$^{-1}$ FW) for the first and second growing seasons were recorded with the application of compost combined with gypsum. In addition, the data showed that the chlorophyll content was significantly increased with foliar application of Mn-NPs and/or Se-NPs. Consequently, the highest values (0.729 and 0.749 mg g$^{-1}$ FW) were obtained with the application of Mn-NPs + Se-NPs. The chlorophyll content was highly significantly increased due to the interaction between all treatments, and the highest values (0.930 and 0.950 mg g$^{-1}$ FW) were recorded with compost + gypsum combined with foliar application of Mn-NPs + Se-NPs in both the growing seasons, as shown in Table 4. The application of nanoparticles to plants improves crop productivity under salinity stress by enhancing plant photosynthesis efficiency and the production of proteins, as mentioned in [31,32].

*3.4. Superoxide Dismutase and Catalase*

As shown in Table 5, that superoxide dismutase (SOD) activity in the plants was highly significantly increased with the application of compost and gypsum or their combination. The highest SOD values (115.71 and 115.13 μg$^{-1}$) were recorded with compost + gypsum for both the growing seasons. Catalase (CAT) also showed the same trend, and the highest values (17.38 and 17.41 μM) were recorded with the application of compost + gypsum for the two growing seasons. The data showed that CAT and SOD were significantly increased with foliar application of Mn-NPs and/or Se-NPs. The highest values of SOD (101.8 and

101.23 $\mu g^{-1}$) and CAT (15.57 and 15.63 $\mu M$) were recorded with the foliar application of Mn-NPs + Se-NPs insalt-affected soils in both seasons. In addition, the SOD and CAT activities were clearly affected by the interaction of all treatments under this study.

**Table 5.** Combined effects of soil amendments and foliar application of some nanoparticles on superoxide dismutase (SOD) and catalase (CAT) of faba beans during the 2020/2021 and 2021/2022 growing seasons.

| Treatments | | Superoxide Dismutase ($\mu g^{-1}$) | | Catalase ($\mu M$) | |
|---|---|---|---|---|---|
| **A** | **B** | **2020/2021** | **2021/2022** | **2020/2021** | **2021/2022** |
| Control | without | 54.57 ± 0.05 [p] | 54.64 ± 0.04 [p] | 8.17 ± 0.06 [o] | 8.34 ± 0.07 [o] |
| | Mn-NPs | 71.60 ± 0.11 [n] | 71.65 ± 0.01 [n] | 9.59 ± 0.05 [m] | 9.61 ± 0.02 [m] |
| | Se-NPs | 78.33 ± 0.10 [l] | 78.51 ± 0.02 [l] | 10.43 ± 0.04 [k] | 10.49 ± 0.01 [k] |
| | Mn + Se | 83.92 ± 0.05 [i] | 84.16 ± 0.03 [i] | 11.34 ± 0.03 [i] | 11.48 ± 0.05 [i] |
| Compost | without | 76.82 ± 0.05 [m] | 77.43 ± 0.03 [m] | 10.26 ± 0.06 [l] | 10.35 ± 0.02 [l] |
| | Mn-NPs | 87.88 ± 0.08 [g] | 88.15 ± 0.07 [g] | 11.65 ± 0.04 [h] | 11.66 ± 0.02 [h] |
| | Se-NPs | 92.63 ± 0.05 [f] | 93.14 ± 0.03 [f] | 12.60 ± 0.01 [f] | 12.65 ± 0.01 [f] |
| | Mn + Se | 95.42 ± 0.05 [d] | 96.13 ± 0.03 [d] | 15.68 ± 0.04 [d] | 15.71 ± 0.01 [d] |
| Gypsum | without | 60.60 ± 0.10 [o] | 61.13 ± 0.03 [o] | 8.70 ± 0.02 [n] | 8.73 ± 0.02 [n] |
| | Mn-NPs | 79.29 ± 0.05 [k] | 80.52 ± 0.04 [k] | 11.07 ± 0.13 [j] | 11.13 ± 0.02 [j] |
| | Se-NPs | 82.47 ± 0.13 [j] | 83.15 ± 0.07 [j] | 11.90 ± 0.03 [g] | 11.95 ± 0.02 [g] |
| | Mn + Se | 86.28 ± 0.15 [h] | 87.15 ± 0.07 [h] | 12.62 ± 0.05 [f] | 12.64 ± 0.02 [f] |
| C + G | without | 93.80 ± 0.04 [e] | 94.18 ± 0.05 [e] | 12.72 ± 0.04 [e] | 12.72 ± 0.02 [e] |
| | Mn-NPs | 115.13 ± 1.00 [b] | 116.47 ± 0.04 [b] | 16.86 ± 0.03 [c] | 16.86 ± 0.03 [c] |
| | Se-NPs | 112.26 ± 1.41 [c] | 112.47 ± 0.06 [c] | 17.32 ± 0.05 [b] | 17.36 ± 0.03 [b] |
| | Mn + Se | 139.30 ± 0.53 [a] | 139.74 ± 0.36 [a] | 22.63 ± 0.04 [a] | 22.71 ± 0.04 [a] |
| Main plot (A) | LSD$_{0.05}$ | 0.006 | 0.006 | 0.005 | 0.007 |
| | LSD$_{0.01}$ | 0.009 | 0.009 | 0.008 | 0.012 |
| Sub plot (B) | LSD$_{0.05}$ | 0.008 | 0.008 | 0.006 | 0.005 |
| | LSD$_{0.01}$ | 0.01 | 0.01 | 0.008 | 0.006 |
| Interaction (A × B) | LSD$_{0.05}$ | 0.01 | 0.016 | 0.012 | 0.009 |
| | LSD$_{0.01}$ | 0.02 | 0.021 | 0.012 | 0.013 |

Means followed by different letters indicate significant differences among treatments according to the Duncan's test ($p < 0.01$). Values are means ± standard deviation (SD) from three replicates (means ± SD).

The highest activities of SOD (139.3 and 139.74 $\mu g^{-1}$) and CAT (22.63 and 22.71 $\mu M$) were recorded due to the interaction between compost + gypsum and foliar application of Mn-NPs + Se-NPs in both the growing seasons. In the present work, three combined treatments were most effective on plant growth insalt-affected soils and enhanced the plant tolerance to salinity. These combination treatments are as follows: gypsum + compost + Mn + Se, followed by compost + Mn + Se, and then gypsum + Mn + Se. Therefore, the outcomes of this study showed that the use of nanomaterials had an important role in alleviating the negative effects of salt stress and was the most effective solution. The results also revealed that under salinity stress, the activities of CAT and SOD enzymes increased in all treatments compared to the control. This may be because the negative impacts of salinity on plant growth were alleviated due to the application of compost and/or Si-NPs and Mn-NPs as observed in [9,25,27].

### 3.5. Seed and Straw Yield of Faba Beans

As shown in Table 6, the seed and straw yields of faba beans were significantly increased with application of compost and/or gypsum. The highest yield values of seed (2076.8 and 2116.6 kg ha$^{-1}$) and straw (3184.7 and 3248.6 kg ha$^{-1}$) were recorded with compost + gypsum for the 2020/2021 and 2021/2022 growing seasons. The data also showed that the seed and straw yields of faba beans were highly significantly increased by foliar application of Mn-NPs and/or Se-NPs in both growing seasons. The highest values of seed (1842.19and 1867.99 kg ha$^{-1}$) and straw (2828.76 and 2934.79 kg ha$^{-1}$) yields were recorded with the foliar application of Mn-NPs + Se-NPs insalt-affected soil. In addition, the interaction of all studied treatments strongly increased seed and straw yields of faba beans. Therefore, the highest values of seed (2544.79 and 2559.19 kg ha$^{-1}$) and straw (3817.20 and 4099.99 kg ha$^{-1}$) yields were achieved with the interaction between soil compost + gypsum and foliar application of Mn-NPs + Se-NPs in the two growing seasons.

These results may be due to the significant and effective role of compost and gypsum in improving the physical and chemical properties of the soil, which led to an increase in the ability of the plant to absorb water and nutrients and thus increased the rate of metabolism and the chlorophyll and proline contents. Foliar application with nanoparticles also had a positive effect in increasing the ability of the plant to overcome stress conditions. Selenium is a beneficial element for plants and has a biostimulant effect as photocatalysis and plant growth increase plant metabolism, crop quality, and stress tolerance. These results are supported by [52]. The combination of adding organic conditioners and gypsum and spraying nanoparticles led to a significant improvement in the grain and straw yields. These results are supported by [3,18,20–24].

**Table 6.** Effect of the application of selected soil amendments and nanoparticles on seed and straw yields of faba beans (kg ha$^{-1}$) during the 2010/2021 and 2021/2022growing seasons.

| Treat. | | Seed | | Straw | |
|---|---|---|---|---|---|
| **A** | **B** | **2020/2021** | **2021/2022** | **2020/2021** | **2021/2022** |
| Control | without | 770.40 ± 2.00 [p] | 803.18 ± 1.15 [o] | 1232.64 ± 3.20 [o] | 844.80 ± 3.46 [P] |
| | Mn-NPs | 890.40 ± 3.05 [o] | 917.59 ± 4.16 [n] | 1418.28 ± 4.85 [n] | 1466.40 ± 2.00 [o] |
| | Se-NPs | 938.40 ± 2.00 [n] | 984.00 ± 2.00 [m] | 1492.06 ± 3.18 [m] | 1562.40 ± 2.00 [n] |
| | Mn + Se | 986.40 ± 2.00 [m] | 1022.40 ± 3.05 [l] | 1568.38 ± 3.18 [l] | 1627.2 ± 6.00 [m] |
| Compost | without | 1303.92 ± 3.00 [l] | 1358.40 ± 2.00 [k] | 2047.27 ± 4.79 [k] | 2135.18 ± 3.05 [l] |
| | Mn-NPs | 1649.59 ± 4.16 [i] | 1667.04 ± 1.10 [i] | 2573.38 ± 6.49 [i] | 2606.40 ± 2.00 [j] |
| | Se-NPs | 1792.80 ± 2.00 [g] | 1815.19 ± 3.05 [g] | 2778.84 ± 3.10 [g] | 2808.79 ± 3.05 [j] |
| | Mn + Se | 1875.98 ± 3.05 [e] | 1907.18 ± 3.05 [e] | 2889.02 ± 4.70 [e] | 2940.00 ± 2.00 [e] |
| Gypsum | without | 1403.18 ± 1.15 [k] | 1423.99 ± 3.05 [j] | 2203.01 ± 1.81 [j] | 2238.38 ± 55.03 [k] |
| | Mn-NPs | 1707.98 ± 3.05 [h] | 1746.38 ± 3.05 [h] | 2664.48 ± 4.76 [h] | 2724.79 ± 3.05 [h] |
| | Se-NPs | 1839.19 ± 3.05 [f] | 1860.00 ± 2.00 [f] | 2850.74 ± 4.70 [f] | 2882.40 ± 3.46 [f] |
| | Mn + Se | 1961.59 ± 4.16 [d] | 1982.40 ± 2.00 [d] | 3040.46 ± 6.45 [d] | 3072.00 ± 2.00 [d] |
| C + G | without | 1635.98 ± 3.05 [j] | 1667.18 ± 1.15 [i] | 2568.50 ± 4.79 [i] | 2619.24 ± 2.30 [i] |
| | Mn-NPs | 1982.4 ± 3.96 [c] | 2036.78 ± 3.05 [c] | 3072.72 ± 5.36 [c] | 2911.20 ± 3.46 [c] |
| | Se-NPs | 2143.99 ± 3.05 [b] | 2203.20 ± 4.00 [b] | 3280.32 ± 4.67 [b] | 3363.98 ± 6.11 [b] |
| | Mn + Se | 2544.79 ± 1.15 [a] | 2559.19 ± 3.05 [a] | 3817.20 ± 1.73 [a] | 4099.99 ± 3.05 [a] |
| Main plo t(A) | LSD$_{0.05}$ | 1.17 | 1.06 | 1.84 | 1.09 |
| | LSD$_{0.01}$ | 1.78 | 1.6 | 2.78 | 1.65 |
| Sub Its write as it plot (B) | LSD$_{0.05}$ | 1.08 | 0.83 | 1.68 | 1.19 |
| | LSD$_{0.01}$ | 1.47 | 1.12 | 2.28 | 1.62 |
| Interaction (A × B) | LSD$_{0.05}$ | 2.17 | 1.66 | 3.37 | 2.39 |
| | LSD$_{0.01}$ | 2.94 | 2.25 | 4.57 | 3.24 |

Notices: Means followed by different letters indicate significant differences among treatments according to the Duncan's test ($p < 0.01$). Values are means ± standard deviation (SD) from three replicates (means ± SD).

### 3.6. Productivity of Irrigation Water (PIW)

The amount of water irrigation during the bean growing season was calculated as 2880 m$^3$ ha$^{-1}$. As shown in Figure 3, PIW (kg m$^{-3}$) was significantly increased with the application of compost and/or gypsum, and the highest values (0.72 and 0.73 kg m$^{-3}$) were recorded with compost + gypsum for the two growing seasons. These results are supported by the authors of [2], who observed that PIW was significantly increased with the application of soil amendments. In addition, the data showed that PIW was highly significantly increased with foliar application of Mn-NPs and/or Se-NPs. The highest values (0.64 and 0.65 kg m$^{-3}$) were obtained with the application of Mn-NPs + Se-NPs compared to the control(0.44 and 0.46 kg m$^{-3}$) for the 2020/2021 and 2021/2022 growing seasons. On the other hand, PIW values were clearly increased by the interaction of all treatments. The highest values of PIW (0.88 and 0.89 kg m$^{-3}$) for the 2020/2021 and 2021/2022 growing season were recorded with the interaction between compost and gypsum as well as foliar application of Mn-NPsand Se-NPs. Increasing water productivity is a primary goal in modern agriculture, and it is necessary to maintain food security and agricultural sustainability. These results are supported by the authors of [31,32], who observed that the addition of nanofertilizers to plants grown under salinity stress improved water use efficiency.

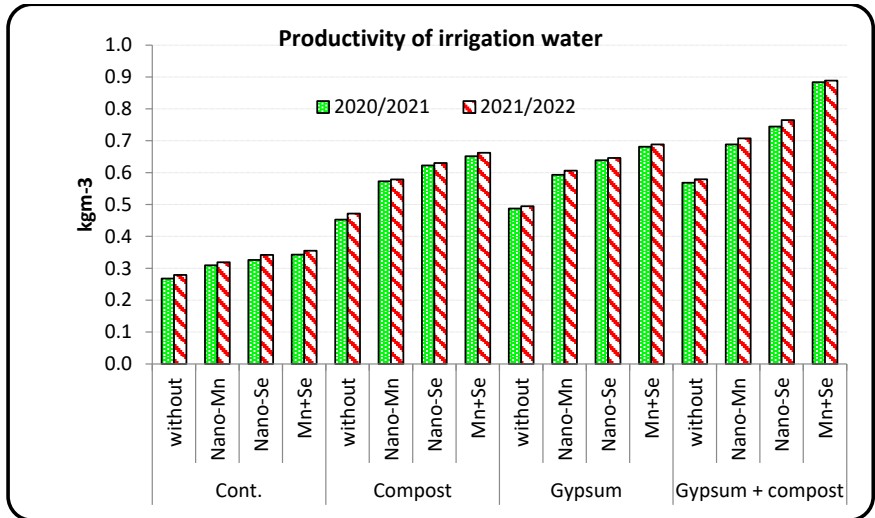

**Figure 3.** Effect of the application of soil amendments and foliar application of some nanoparticles on the productivity of irrigation water (PIW, kg seed m$^{-3}$) during the 2020/2021 and 2021/2022 growing seasons.

## 4. Conclusions

The results of this study showed thatthe application of compost and gypsum significantly decreased soil salinity (EC), soil sodicity (ESP), soil bulk density (BD), and soil penetration resistance (SPRa) and significantly increased soil porosity and soil basic infiltration (IR). In addition, foliar application of nanomaterials improved plant salt tolerance. In addition, significant positive effects onthe 100-grain weight and proline, chlorophyll, superoxide dismutase, and catalase contents were observed due to the interaction between gypsum + compost and Mn-NPs + Se-NPs, which enhanced both the seed and straw yields of faba beans compared to the alternative treatments. Therefore, it might be inferred that using gypsum + compost and foliar application of Mn-NPs + Se-NPs may be a key strategy for improving some chemical and physical properties of soil and the yield and water productivity of faba beans in salt-affected soils.

**Author Contributions:** Conceptualization, E.M.S.; methodology, M.M.A.; software, M.M.A. and A.A.H.; validation, H.M.A.; formal analysis, H.M.A., E.M.S. and A.A.H.; investigation, M.M.A. and E.M.S.; resources, H.M.A.; data curation, H.M.A.; writing—original draft, H.M.A. and E.M.S.; writing—review and editing, M.M.A. and A.A.H. All authors have read and agreed to the published version of the manuscript.

**Funding:** This research received no external funding.

**Data Availability Statement:** Not applicable.

**Conflicts of Interest:** The authors declare no conflict of interest.

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
