# Peer review of "Effect of Gypsum, Compost, and Foliar Application of Some Nanoparticles in Improving Some Chemical and Physical Properties of Soil and the Yield and Water Productivity of Faba Beans in Salt-Affected Soils"

_agronomy, doi:10.3390/agronomy13041052_

Round 1

Reviewer 1 Report

Add conclusion lines at the end of abstract

remove abbreviation from the keywords list.

page 2 line : (OM) [6,7].Compos enter space

page 2 citation 12, add scientist name

page citation 13 and 14,, f Faba bean[13, 14].Also, the enter space remove typo

page 2: N, P, K, and Zn . use full form at first use

there so many minor mistake, enter space between sentense

resubmit manuscript with line number, i will review manuscript again

Interesting study.

Author Response

Dear Prof.

a great thanks for your comments

all comments were don in the file

Reviewer 2 Report

Hello Dear,

Good Work.

The manuscript needs a lot of improvements. 

Please let a native English speaker/writer goes through the manuscript.

Please consult with a statistician about your analysis.

Please be consistent even with the author's information "Dept. or Department" "USA, and"

I attached some of my comments.

Best

Author Response

Dear prof

Great thanks for your comments

all of its were done in the file

Round 2

Reviewer 1 Report

I have some minor comments, honestly, without line number very difficult to review the manuscript.

before and after full stop, enter space in each sentence, remove typo

at the end of introduction, clearly write the aims of the study, and objective of this study add into points, easy to understrand for readers

How you optimize the level of these treatment, provide suitable reference

all abbervation and full name use at first use, go through the whole manuscript

page 5 first line remove typo in the enzymes unit

check figure 1A. stand error is cut from the top of figure,, increase y axis and modify the figure

infuture, never submit manuscript without line number.

Author Response

Dear Prof. Attached is a file containing all the responses to your comments
a great thanks

Reviewer 2 Report

I do not think my comments were addressed as I expected. 

Author Response

Point 1: English language Response 1: Dear Prof The manuscript was reviewed by a colleague, Dr. Ahmed Hashem, who works at Agricultural systems technology Department, College of Agriculture, Arkansas State University, Jonesboro, Arkansas, USA, and then reviewed by his professor
Point 2: comments
Response 2: All of your comments have been taken into account as they are useful to enrich the research. Please accept my sincere thanks to you
